# Quantification and Simulation of Landscape Anthropization around the Mining Agglomerations of Southeastern Katanga (DR Congo) between 1979 and 2090

Héritier Khoji Muteya [1,2,*], Dieu-Donné N'Tambwe Nghonda [1,2], François Malaisse [2], Salomon Waselin [2,3], Kouagou Raoul Sambiéni [4], Sylvestre Cabala Kaleba [1], François Munyemba Kankumbi [1], Jean-François Bastin [2], Jan Bogaert [2,4] and Yannick Useni Sikuzani [1]

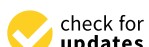



1 Unité Ecologie, Restauration Ecologique et Paysage, Faculté des Sciences Agronomiques, Université de Lubumbashi, Lubumbashi BP 1825, Democratic Republic of the Congo; nghondan@unilu.ac.cd (D.-D.N.N.); kalebac@unilu.ac.cd (S.C.K.); kankumbim@unilu.ac.cd (F.M.K.); sikuzaniu@unilu.ac.cd (Y.U.S.)
2 Unité Biodiversité et Paysage, Université de Liège—Gembloux Agro-BioTech, 5030 Gembloux, Belgium; malaisse1234@gmail.com (F.M.); w.salomon@uliege.be (S.W.); jfbastin@uliege.be (J.-F.B.); j.bogaert@uliege.be (J.B.)
3 Campus Henri Christophe de Limonade, Université d'Etat d'Haïti, 1130, Rte Nationale # 6 Limonade, Limonade HT 1130, Haiti
4 Ecole Régionale Post-Universitaire d'Aménagement et de Gestion Intégrés des Forêts et Territoires Tropicaux (ERAIFT), Université de Kinshasa (UNIKIN), Kinshasa BP 15373, Democratic Republic of the Congo; krsambieni@uliege.be
* Correspondence: hkhoji@doct.uliege.be; Tel.: +243-9980-02204

**Abstract:** In Southeastern Katanga, mining activities are (in)directly responsible for deforestation, ecosystem degradation and unplanned building densification. However, little is known about these dynamics at the local level. First, we quantify the landscape anthropization around four agglomerations of Southeastern Katanga (Lubumbashi, Likasi, Fungurume and Kolwezi) in order to assess the applicability of the Nature–Agriculture-Urbanization model based on the fact that natural landscapes are replaced by anthropogenic landscapes, first dominated by agricultural production, and then built-up areas. Secondly, we predict evolutionary trends of landscape anthropization by 2090 through the first-order Markov chain. Mapping coupled with landscape ecology analysis tools revealed that the natural cover that dominated the landscape in 1979 lost more than 60% of its area in 41 years (1979–2020) around these agglomerations in favor of agricultural and energy production, the new landscape matrix in 2020, but also built-up areas. These disturbances, amplified between 2010 and 2020, are more significant around Lubumbashi and Kolwezi agglomerations. Built-up areas which spread progressively will become the dominant process by 2060 in Lubumbashi and by 2075 in Kolwezi. Our results confirm the applicability of the Nature–Agriculture-Urbanization model to the tropical context and underline the urgency to put in place a territorial development plan and alternatives regarding the use of charcoal as a main energy source in order to decrease the pressure on natural ecosystems, particularly in peri-urban areas.

**Keywords:** Southeastern Katanga; mining agglomeration; landscape anthropization; land cover change

## 1. Introduction

Tropical forests are one of the richest environments in terms of biodiversity [1]. Consequently, forest ecosystems are recognized as the key pillar in maintaining the overall balance of the planet [2] since human populations depend on and are linked to these ecosystems for their well-being [3]. However, in an effort to increase economic development, changes in anthropogenic activities over the past 50 years have caused significant pressures on forest ecosystems, threatening their ability to provide ecosystem services [4].

Indeed, besides natural causes such as forest fires, diseases or pests that can affect trees, anthropogenic pressure on forest ecosystems is the main cause of changing their spatial pattern [5]. The situation is most alarming in sub-Saharan Africa, where forest ecosystems are experiencing rapid degradation [4]. Countries in this region of the world already have higher proportions than European countries in terms of pressure on land use, changes in atmospheric composition, disruption of the water cycle, alteration of the nitrogen cycle and loss of biodiversity per unit of wealth produced [6]. For example, in sub-Saharan Africa, the urban population is expected to increase five-fold, from 200 million people in 2000 to one billion in 2050, while the urbanized area will increase 12-fold over the same period, from $\pm 26,500$ km$^2$ to $\pm 325,500$ km$^2$ [7].

The Democratic Republic of the Congo (DRC) is a country with remarkable forest resources [8] that is no exception to this trend. It is among the countries with one of the highest population growth rates. Its population was estimated at 30.7 million in 1984, compared to 81.3 million in 2017 [9]. The expansion of survival activities, among other shifting cultivation and charcoal production, are the major causes of deforestation and forest degradation, concentrated mainly around urban areas [10,11]. With less than 10% of the Congolese population having access to electricity, the country is among those with the lowest electrification rate [12]. Thus, as an illustration, an annual deforestation rate of 0.23% was noted between 2000 and 2010, corresponding to a loss of 371,180 km$^2$ of forest per year over the entire country to meet the energy demand [10]. It should be noted that this deforestation concerns primary and secondary forests in the central basin [8]. This deforestation is mainly concentrated in the provinces of Kinshasa and Bas-Congo, in the east of the DRC and around the medium-sized cities along the Congo River [13] but also in Southeastern Katanga, where miombo woodland is disappearing in favor of anthropogenic land cover [14].

The landscapes of Southeastern Katanga have been profoundly modified by the mining activities developed since 1909, which have led to agricultural development to feed the workers of the mining companies and the creation of new agglomerations for their homes [15]. The consequences of the economic attractiveness of the mining industry, especially since the liberalization of the mining sector in 2002, are a massive rural exodus and rapid population growth [16,17]. The resulting rapid spatial expansion of urban areas is likely to induce a range of negative socio-economic and environmental impacts, such as loss of agricultural land, deforestation, persistent land insecurity, inadequate amenities and degradation of ecosystem services [18]. Furthermore, urbanization contributes to the creation of single ecosystems through biotic homogenization [19]. The future of vegetation is thus largely dependent on the spatial pattern of Southeastern Katanga cities, which are constantly expanding their hold on areas that still retain their rural character, leading to the creation of peri-urban areas [17,20]. If peri-urban areas are considered land reserves for the extension of cities, their dynamics and management are not planned in Southeastern Katanga [20,21]. The demographic pressure on urban areas and subsequently on peri-urban areas will continue, and without a minimum of organization, environmental degradation could accelerate and access to land for housing and agriculture would become increasingly competitive for the population [22].

As human influence on these landscapes has increased dramatically, methods are needed to properly monitor and evaluate changes in land use and land cover [23]. Analysis of the spatiotemporal dynamics of natural landscapes is important for monitoring and quantifying the consequences of human activities [24]. For this reason, the quantification of landscape spatial patterns and their evolution over time is becoming a major concern due to the magnitude of change [25] and will increase in the coming decades. Among the different existing approaches, the landscape evolution of the model of Ref [26] based on the fact that natural landscapes are replaced by anthropogenic landscapes, first dominated by agricultural production and then urbanization [27], has been proposed. It must be noted that this model, designed for landscapes in temperate regions, has not yet been tested on landscapes in developing countries.

For this reason, we quantify the landscape anthropization around the agglomerations of Southeastern Katanga. The specific objectives are (i) to evaluate the applicability of the evolutionary sequence model of natural landscapes of the model of Ref [26] in a tropical mining context and (ii) to determine the evolutionary trends of landscape anthropization by 2090. We hypothetized that the natural cover undergoes a regressive dynamic in Southeastern Katanga to the benefit of agricultural and energy production. In the lack of an adequate planning policy, this trend will continue in the future, mainly in favor of building densification and mining coverage. The model of [26] would be applicable in the miombo woodland context when combining agricultural production with energy production, two of the important factors leading to Savanna development [28].

## 2. Materials and Methods

### 2.1. Research Area

The present study was conducted in four agglomerations (Lubumbashi: 11°35.732′–11°38.227′ S and 27°18.158′–27°49.16′ E; Likasi: 11°1.474′–11°4.437′ S and 26°35.842′–27°6.37′ E; Kolwezi: 10°35.735′–10°39.026′ and 25°7.945′ S–25°35.418′ E; Fungurume: 10°39.137′–10°41.359′ S and 25°58.870′–26°29.322′ E) (Figure 1). These agglomerations are known as the main poles of landscape anthropization in Southeastern Katanga [29–31]. With sandy soil in the north and sandy-clay soil in the south, the relief of the Southeastern Katanga is characterized by a succession of mineralized and non-mineralized hills, mountain ranges, plateaus, plains and valleys [32]. This region belongs to the Cw climatic type according to Köppen's classification, with a rainy season (from November to April) and a dry season for the rest of the year [33]. The average annual temperature was 20 °C in the second half of the 20th century, but ongoing warming in the region is evident [34]. Total annual rainfall varies from 1200 mm (in the Lubumbashi area) to nearly 1600 mm (in the Kolwezi area), according to [35]. This pattern remains valid, although recent studies suggest a trend toward a later onset of rains and lower average annual rainfall [36]. Southeastern Katanga is a subregion of the greater Sudan-Zambezian region, otherwise lying within the Zambezian domain, characterized primarily by humid miombo woodland dominated by a few species, mostly from the genera *Brachystegia*, *Julbernardia* and *Isoberlinia* [37], miombo woodland dominated by *Marquesia* and the high termite mounds of miombo woodland alongside savannas [38]. The main economic activities in Southeastern Katanga are mining, agriculture, livestock, trade (mostly informal) and services for a population of more than 2 million inhabitants in Lubumbashi, nearly 800,000 inhabitants in Likasi and Kolwezi, compared to nearly 400,000 inhabitants in Fungurume [39,40].

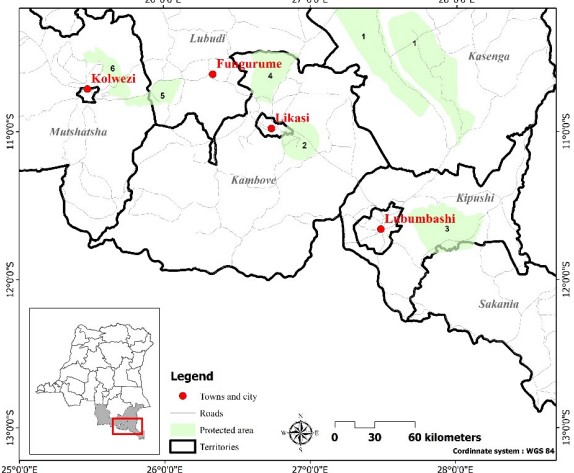

**Figure 1.** Location of study sites in Southeastern Katanga: Lubumbashi, Likasi, Kolwezi and Fungurume. These agglomerations are located near protected areas (Kundelungu National Park (1),

Lufira Biosphere Reserve (2), hunting domains such as Kiziba-Baluba (3), Mufufya (4), Basse Kando (5), Tshangalele (6)). Lubumbashi and Likasi are located in Upper-Katanga province, respectively, in Kipushi and Kambove territories, while Kolwezi and Fungurume are located in Lualaba province, respectively, in Mutshatsha and Lubudi territories.

### 2.2. Fitting the Nature–Agriculture-Urbanisation model to the Land-Cover of Southeastern Katanga

#### 2.2.1. Data

The 4 poles of anthropization were isolated from 9 Landsat images covering the period between 1979 and 2020. These Landsat images were acquired from MSS (1979, 1984) with 60 m resolution, TM (1990, 1995) with 30 m resolution, ETM+ (2000, 2005, 2010) with 30 m resolution and OLI-TIRS (2015 and 2020) with 30 m resolution, depending on their availability. Lubumbashi and Likasi cities were isolated on the same scene (Path/Row: 173/68) and those of Kolwezi and Fungurume on another scene (Path/Row: 174/67). These Landsat images were acquired during the dry season (June–August) to maintain consistency in the spectral response of the different vegetation covers [41]. Overall, the Landsat images, which were used in this work, cover two distinct periods: the period before the liberalization of the mining sector (1979, 1984, 1990, 1995 and 2000) and the period after the liberalization of the mining sector (2005, 2010, 2015 and 2020) in the Democratic Republic of Congo. The last period was characterized by a significant population growth due to the rural exodus favored by the expansion of mining activities [16]. The period before mining liberalization was also characterized by the civil wars of 1980 and the Sovereign National Conference between 1990 and 1992, while the post-liberalization period was characterized by the economic crisis of 2008.

#### 2.2.2. Preprocessing of Landsat Images

The Landsat images were georeferenced in the UTM Zone 35S datum based on the WGS 84 reference ellipsoid. The pixel size was homogenized to 30 m resolution by applying cubic resampling that preserves the radiometric values of the images [42]. Furthermore, 1979, 1984, 1990, 1995, 2000, 2005, 2010 and 2015 Landsat images were orthorectified with reference to the 2020 Landsat image and the geometric accuracy of the calibration was acceptable for change analysis. Finally, the calibration based on the subtraction of dark objects was used for the radiometric correction because it allows us to remove atmospheric measurements [43].

#### 2.2.3. Classification of Landsat Images

The false color composite of the selected Landsat images was performed by combining the near-infrared, red and blue channels, with the first two channels providing the best discrimination of vegetation [44]. Then, for each land cover type on the false color composite, training areas were isolated on the composite images and located on the ground using a GPS (GARMIN 64S, accuracy ±3 m). Twenty homogeneous training areas selected for each land cover type were used to perform the supervised classification of Landsat images based on the maximum likelihood algorithm. This algorithm uses the statistics of the training areas to calculate the maximum probability of pixels belonging to predefined land cover [45]. Six land cover types were first discriminated, namely forest (open forest, dense, dry forest and gallery forest), savannahs (tree and shrub savannahs), fields and fallows, bare soil and built-up area, wetland and water.

The quality of the Landsat images classification was assessed through the overall accuracy, which characterizes the proportion of well-classified pixels, and the Kappa coefficient, which corresponds to the ratio between correctly classified pixels and the set of pixels considered [46]. The 6 previous land cover types were grouped into 3 land cover types to fit the model of Ref [26]: natural cover (forest, swamps), built-up area (built complex and bare soil, peri-urban landscape, mines included) and agricultural production to which energy production (savannas, fields, and fallows) was added. In an African context, we

associated the peri-urban zone with the urban zone because these two zones intermingle to the point of disaggregating the boundaries that previously separated them [42]. According to [47], an urban zone is then defined as a zone where the built surfaces are dominant or where buildings are present as continuous pattern elements. Peri-urbanization refers to a dynamic process that consists mainly of the expansion of artificial areas into natural, semi-natural and agricultural areas [48].

Around each study site, referred to in this work as a "zone," an anthropization radius of 25 km, corresponding to an area of 1963.5 km$^2$, was placed from its telecommunication station, considered the center of these zones. This radius was chosen to reduce overlap between closely spaced sites (Figure 2). All analyses were conducted using ArcGIS 10.8 software.

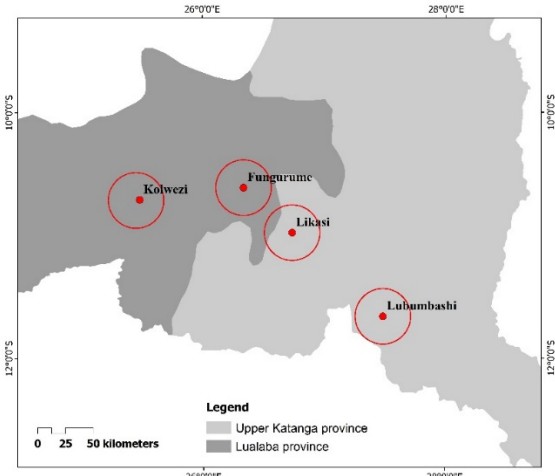

**Figure 2.** Measuring the anthropization of the landscape around the main agglomerations in Southeastern Katanga. The circle around the 4 poles of anthropization corresponds to the part of the territory studied and is referred to in this study as the "zone".

### 2.2.4. Simulation of the Landscape Dynamics by the 1st Order Markov Chain

The MARKOV module of the IDRISI 17.0 software was used to highlight the first transition probability matrix from 1979 to 2020 for each zone selected in this study. These transition probability matrices (1979–2020) quantify the extent of transfer between land cover types and allows the prediction of future changes [49]. The second matrix calculated by the software is the area transition matrix (data in a number of pixels). This matrix is directly related to the transition probability matrix and derived from the multiplication of the probabilities by the total number of pixels of the most recent map for each land cover. The 2020 land cover image was used as the starting point for the change simulation. The suitability (constraint) maps were constructed for each land cover class using the Multi-Criteria Evaluation module. From these transition probability matrices, the CA MARKOV module was used to predict land cover changes in 2030, 2040, 2050, 2060, 2070, 2080 and 2090, the year from which the matrices remained unchanged [50].

The validation of the reliability of the simulation of Markov (first-order) performed was carried out using the Chi-square ($\chi$2) statistical test for a threshold value of 5% in order to compare observed (1979–2000) and expected values [30,51].

### 2.2.5. Analysis of Landscape Anthropization Dynamics

The extent of landscape anthropization was assessed from the calculation of the proportion of land cover in the landscape (PLAND) [52]. A decrease in the PLAND may suggest the fragmentation or disappearance of its patches. Finally, the disturbance index (U), defined as the ratio of the cumulative area of anthropogenic land cover (agricultural and energy production, built-up area) in the landscape to the cumulative area of natural land cover [53], was calculated to quantify the level of natural landscape degradation

over time. To test the year and zone effect, the evaluation of the normal distribution fit of the data was performed through the Shapiro and Wilk test. Subsequently, the analysis of variance (ANOVA), with a probability threshold of 5%, was used to evaluate the zone effect and the year effect on the landscape anthropization evolution in the Southeastern Katanga. The comparison of means was performed using Tukey's DSH test. These statistics were generated using R 4.0.5 software.

## 3. Results

### 3.1. Analysis of Satellite Data: Classification and Mapping (1979 to 2020)

The accuracy of the supervised classification of the Landsat images covering the zones of Lubumbashi and Likasi, as well as those of Kolwezi and Fungurume, indicates overall accuracy values between 77.8% and 93.9%, and those of the Kappa between 73.8% and 92.2% (Table 1). These values suggest, in general, that the discrimination between land cover types is statistically reliable according to [54].

**Table 1.** Accuracy of supervised classification of Landsat images of 1979, 1984, 1990, 1995, 2000, 2005, 2010, 2015 and 2020 supported by the maximum likelihood algorithm.

| Year | Landsat Scene Lubumbashi-Likasi | | Landsat Scene Kolwezi—Fungurume | |
|---|---|---|---|---|
| | Overall Accuracy (%) | Kappa (%) | Overall Accuracy (%) | Kappa (%) |
| 1979 | 91.7 | 90.1 | 90.8 | 89.1 |
| 1984 | 93.9 | 92.2 | 87.4 | 85.5 |
| 1990 | 91.5 | 89.7 | 84.6 | 83.3 |
| 1995 | 88.2 | 85.6 | 82.0 | 79.7 |
| 2000 | 77.9 | 76.4 | 92.2 | 90.1 |
| 2005 | 88.3 | 85.0 | 91.5 | 89.2 |
| 2010 | 84.5 | 83.1 | 91.6 | 88.6 |
| 2015 | 80.7 | 73.8 | 88.7 | 84.4 |
| 2020 | 86.2 | 81.9 | 93.1 | 91.8 |

The visual analysis of the land cover maps shows, for the four zones, spatial changes in the landscape between 1979 and 2020 materialized by a regression of the natural cover that was replaced by agricultural and energy production as well as built-up area that experienced a progressive spatial dynamic in the landscape (Figure 3). The "water" cover was excluded for further analysis because of its static nature in Likasi and Kolwezi zones, but also because of its relatively small proportion in Lubumbashi and Fungurume zones.

According to the first-order Markov model, its validation carried out on the basis of the $\chi^2$ test of comparison of probabilities and simulated in 2020 reveals a non-significant difference, with values of $\chi^2 = 3.2$, $\chi^2 = 3.19$, $\chi^2 = 7.8$ and $\chi^2 = 4.7$, respectively, for the four zones ($p > 0.05$). These results attest that the landscape dynamics of the study areas can be modeled using a first-order Markov chain. The visual analysis of the simulated land cover maps shows a progression of built-up area by 2090, especially in the Lubumbashi and Kolwezi zones, at the expense of other land covers. The Likasi and Fungurume zones show a trend whereby the landscape matrix will still be characterized by consistent coverage of agricultural and energy production (Figure 3).

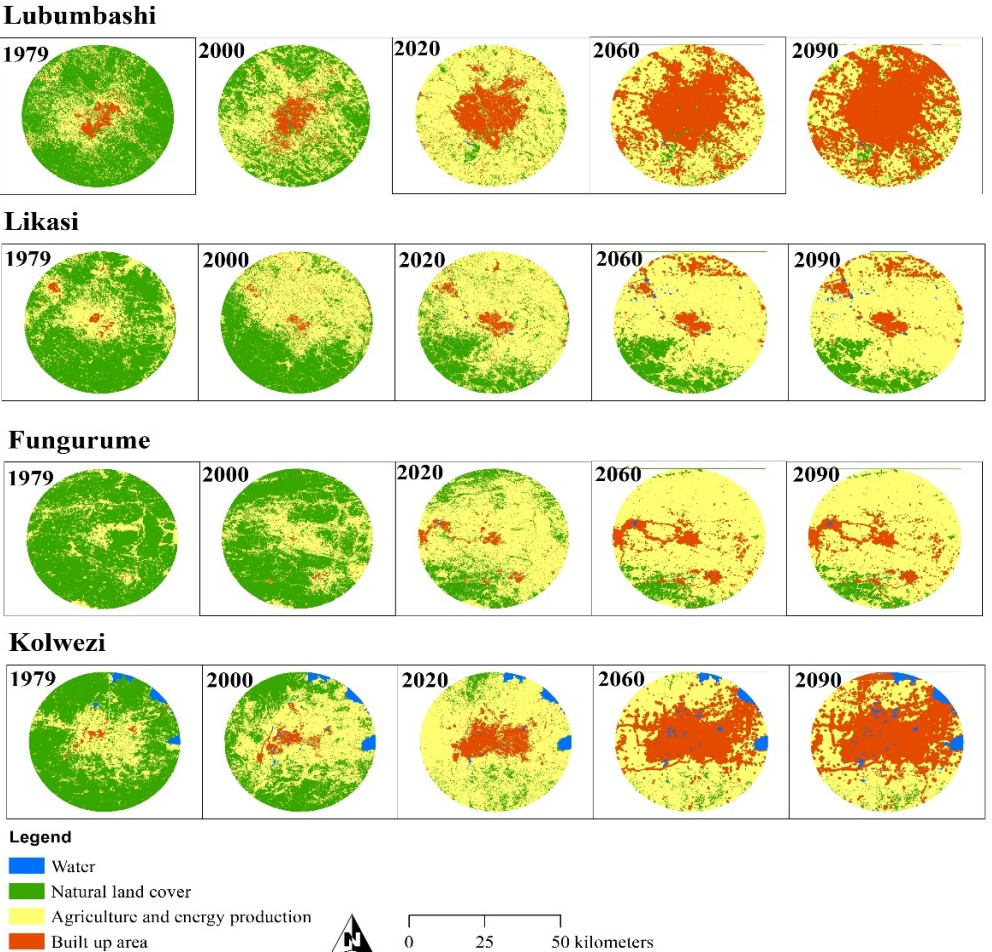

**Figure 3.** Land cover maps of Lubumbashi, Likasi, Fungurume and Kolwezi zones from the supervised classification of Landsat images from 1979, 2000 and 2020 based on the maximum likelihood algorithm as well as the 2060 and 2090 maps from the 1st order Markov chain modelling. The discarded dates do not show much change compared to the nearby posted dates.

### 3.2. Dynamics of Landscape Anthropization between 1979 and 2090

The change in land cover observed between 1979 and 2020 generally indicates the regression of natural land cover in favor of anthropogenic land cover in the four zones studied (Figure 4). In the Lubumbashi zone, the natural land cover that constituted the landscape matrix between 1979 and 1995 experienced a decrease in its proportion over time (−88.5% between 1979 and 2020 or 2.2% annually). In contrast, agricultural and energy production and also built-up areas experienced an expansive dynamic. By tripling its area in 41 years, agricultural and energy production became the landscape matrix. By 2090, Lubumbashi will register a decline of 49.7% of its natural land cover from 2020 and 52% of agricultural and energy production to the benefit of the built-up area, which will be tripled during the same period, thus becoming the landscape matrix for 2060 (Figure 4). These trends are confirmed by the disturbance index, which increased from 0.45 in 1979 to 11.7 in 2020, suggesting that the level of disturbance increased 26-fold in 41 years. The magnitude of disturbance was more remarkable between 2010 and 2020. By 2090, this index will increase 52-fold (Figure 5).

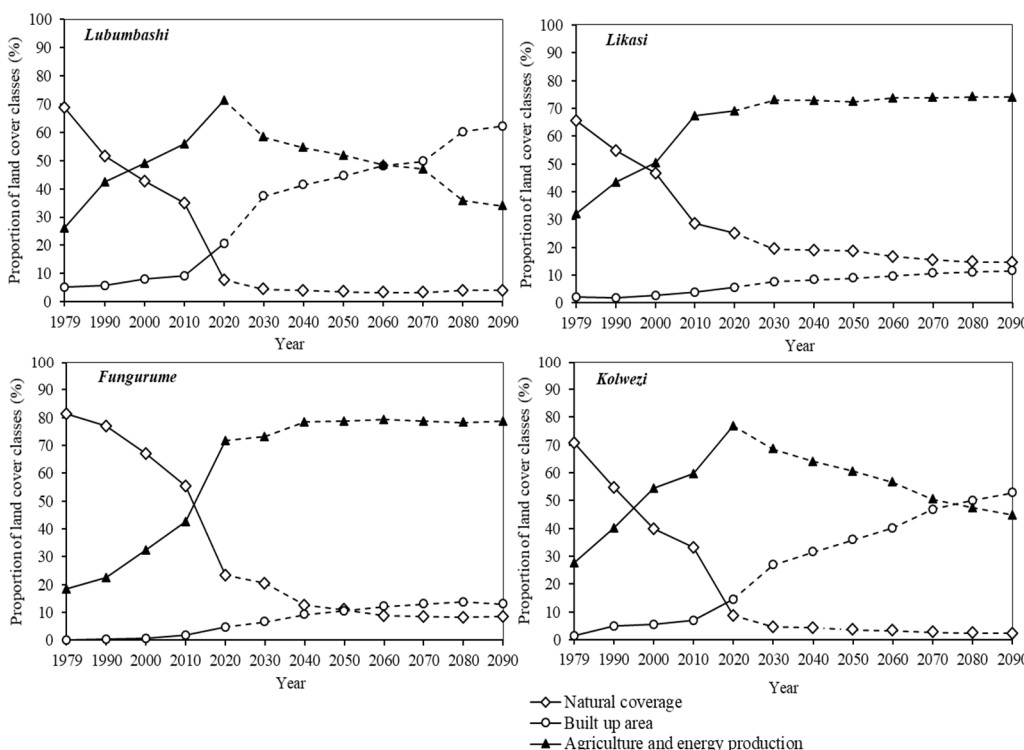

**Figure 4.** Evolution of the total area of the occupation classes between 1979 and 2090 (total value of each anthropization pole = 1963.5 km$^2$) in the Lubumbashi, Likasi, Fungurume and Kolwezi areas.

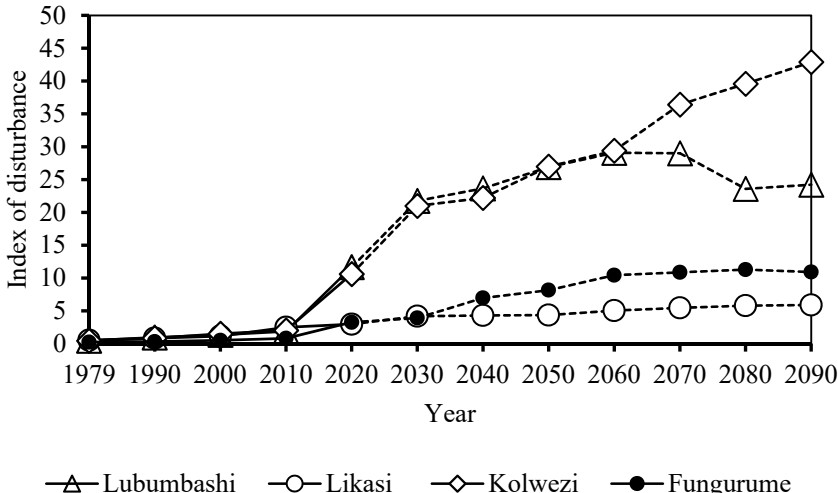

**Figure 5.** Evolution of the disturbance index in the 4 zones of Southeastern Katanga (Lubumbashi, Likasi, Fungurume and Kolwezi) between 1979 and 2090.

The natural land cover, which dominated the landscape of the Likasi zone between 1979 and 1995, decreased by 61.5% (1.5% annually) of its initial area between 1979 and 2020. During the same period, built-up areas as well as agricultural and energy production tripled and doubled their proportions, respectively. Agricultural and energy production currently dominate the landscape since 2000 (Figure 4). This trend will continue until 2090 when the built-up area will double its acreage, while natural land cover will decline by 42.8% and agricultural and energy production will increase slightly (1.06%) (Figure 4). The disturbance index, which increased from 0.52 in 1979 to 2.96 in 2020 and 5.9 in 2090, attests to the fact that the level of landscape disturbance has increased five-fold in 41 years and will be multiplied by 11 in the future (Figure 5).

In the Kolwezi zone, the landscape was dominated by natural land cover between 1979 and 1990 before regressing to 87.8% in 2020 (2.1% annually) in favor of anthropogenic land cover. Built-up area increased nine-fold while agricultural and energy production tripled its proportion between 1979 and 2020 and became the new landscape matrix from 1995 onwards. By 2090, the Kolwezi zone will lose 74% of its natural land cover and 42.6% of its agricultural and energy production to the benefit of built-up areas, which will quadruple its area and will become the landscape matrix from 2075 (Figure 4). The extent of landscape disturbance has increased 26-fold in 41 years, from 0.41 in 1979 to 10.6 in 2020, and will increase 97-fold by 2090 (Figure 5).

The natural land cover that constituted the landscape matrix of Fungurume between 1979 and 2010 lost 71% of its proportion between 1979 and 2020 (1.7% annually). During the same period, built-up areas as well as agricultural and energy production have been multiplied by 24 and 4, respectively. Agricultural and energy production became the new landscape matrix. The same tendency will be observed until 2090 (Figure 4). Natural cover will lose 64.3% of its proportion while built-up areas will triple their acreage. Agricultural and energy production will increase slightly (1.09%) and will remain the matrix in the landscape (Figure 4). These results are confirmed by the evolutionary trend of the disturbance index, whose values have increased from 0.23 in 1979 to 3.25 in 2020, suggesting that in 41 years, the proportion of the anthropogenic land cover in the landscape has increased by a factor of 14 to the expense of the natural class. By 2090, this index will be multiplied by 49 (Figure 5).

Overall, a degradation of the landscape is noted between 1979 and 2020, materialized by the loss of natural cover in the four zones. This loss of natural cover was found to be too low in 1979, low between 1984 and 2000, medium in 2005 and 2010 and accelerated in 2015 and especially in 2020 (df = 8; F = 11.6; $p < 0.001$). This trend was observed in the four zones without significant difference (df = 3; F = 2.1; $p > 0.05$). The opposite situation was observed for built-up areas, which increased remarkably during the same period and more markedly in Lubumbashi, followed by the Kolwezi region, compared to Fungurume and Likasi regions (df = 3; F = 15.02; $p < 0.001$). Agricultural and energy production increased significantly more in the four zones (df = 3, F = 1.7; $p > 0.05$). Its proportion was very low between 1979 and 2000, average between 2000 and 2010 and accelerated between 2010 and 2020 (df = 8; F = 12.1; $p < 0.001$).

In terms of the evolutionary stages of landscape anthropization according to the model of Ref [26], natural cover dominated the landscape during the period before 1995 for Kolwezi, 2000 for Lubumbashi and Likasi and 2015 for Fungurume. This land cover has undergone significant pressures related to anthropogenic activities that have led to a continuous loss of its acreage. Consequently, it loses its status of landscape matrix from the beginning of 2000 for the four zones, the period which coincides with the liberalization of the mining sector in the DRC. The agricultural and energy production patches increased in proportion in the landscape until they become the new matrix and the final trend shows an increase in the built-up area. The patches associated with built-up area development have been steadily expanding in the landscape of the four zones (Figure 4). Between 2020 and 2090, the prediction indicates that this trend will go on until the 2060 and 2075 horizon for Lubumbashi and Kolwezi zones, respectively, but after these years, their landscapes will be dominated by built-up areas. Conversely, for Likasi and Fungurume zones, the trend will not change by 2090 despite the constant expansion of built-up patches in both areas.

## 4. Discussion

### 4.1. Methodological Approach

Landscape dynamics are the result of interactions between societies and their living environment. Thus, the resulting pattern change in space and time [25]. Therefore, through the use of Landsat images acquired at different dates, a study covering a period of more than 30 years is sufficient to observe profound changes, especially in human-dominated landscapes [43,55]. A time step of 5 to 6 years (less than 10 years) between images is

acceptable to analyze the average rate of land cover change over areas undergoing rapid change [25].

Landsat images preprocessing, land cover selection and merging and supervised classification, whose overall accuracies and Kappa coefficient were clearly acceptable in this study, confirm the importance of knowing the study area to improve the quality of image processing despite their coarse resolution [56]. The three land cover types targeted in this study correspond to the main land cover in the Katanga region, for which the determination of their proportion proved to be relevant in order to identify the evolutionary trends of the land cover [52]. The proportion (index) made it possible to analyze the landscape dynamics at the scale of the Katangan Copper belt area [30]. In addition, anthropogenic activities lead to landscape degradation, the extent of which can be quantified through the disturbance index, which has the advantage of being based on the area of land cover and its typology [57].

The Bogaert and others model [26], applied in this study, is a tool used to analyze the historical perspective of typical landscape dynamics. However, it is a model designed for northern landscapes; hence its adaptation to the regional context led to the redefinition of land cover types to explain the evolution of landscapes in Southeastern Katanga.

The Markov simulation based on the transition probability matrix allowed the modeling of the observed dynamics to control the landscape change and predict the evolution of the risk [51]. This model was preferred in this study because it is recognized as the most robust, and it can produce trends that can guide decision-making on landscape evolution over time [58] despite the limitations described by [59]. It has been used to simulate the landscape composition in the Katangan Copperbelt in the DR of Congo [60] and in the forest landscape of the eastern Côte d'Ivoire [51].

### 4.2. Dynamics of Landscape Anthropization in Southern Katanga: Drivers, Patterns and Evolutionary Stages

In Southern Katanga, anthropogenic activities are leading to a regression of natural cover in favor of anthropogenic land cover, confirming the landscape dynamics trends observed within the Katangan Copper belt [29,30,61]. Thus, the miombo woodland, the dominant ecosystem in Southeastern Katanga, is steadily regressing as a result of various anthropogenic activities supported by the population explosion in the region. The scale and pace at which humans are using space are leading to remarkable impacts in terms of deforestation, degradation, overexploitation and desertification, especially in this period of rapidly built-up areas and industrial growth [62]. Mining activities, agricultural production and the persistent increase in urban demand for wood energy are the main causes behind the increased pressure on miombo woodland [63,64]. Reference [65] Findings in the Chingola region of Zambia are in line with our results.

Furthermore, our results confirm the high degree of anthropization in southern Katanga that has led to rapid building densification to meet the housing needs of a growing population [31]. However, several authors recognized that this spatial built-up area expansion is unplanned and leads to the significant degradation of wood resources, especially in built-up areas [17,30]. On the other hand, population growth is accompanied by high food and energy needs, whose search for satisfaction leads to a decrease in the abundance of miombo woodland patches, followed by savanization [66], thus modifying the landscape matrix in southern Katanga. In the context of the widespread dysfunction of the national electricity service and the obsolescence of its equipment otherwise not adapted to rapid city expansion [67], charcoal is considered the main source of energy for cooking meals [68]. In this region, as elsewhere in DR Congo and Africa, the wood-energy sector contributes more than 80% of total domestic energy consumption and is also reportedly responsible for more than 90% of total woody removals from forests [69]. Furthermore, in an area where mineral fertilizers are expensive, forest soils are coveted by local farmers, who also abandon them after two to three cropping seasons to open up new ones [70]. This practice leads to the destruction of large areas of forest. In the region, refs. [71,72] noted deforestation resulting

from an increase in the area of agricultural land and a decrease in forested areas in Malawi and Zimbabwe, respectively.

The rates of regression for natural cover have exceeded 60%, but the extent differs by zone. For illustration, Lubumbashi, which is the second most important city in DR Congo after the capital Kinshasa, both demographically and economically [73], has experienced the highest rate of regression of natural cover (88.5% in 41 years). Despite moving beyond industrial dependence to also become an administrative city and political center [63], Lubumbashi is experiencing high population growth rates and unplanned built-up area growth [74]. However, Kolwezi recorded a similarly high rate of natural cover regression (87.8%), despite the fact that its population is only 1/3 compared to Lubumbashi [40]. This is explained by the fact that the Kolwezi zone has more mining character and history that has negatively impacted its landscape [63], thus justifying its early anthropization alongside the presence of natural savannas. This anthropization accelerated notably after the scission of the large province of Katanga into four new provinces [41]. This would have led to a significant rural exodus in 2016 and a displacement of populations from other areas for artisanal mining in the Kolwezi zone. Lubumbashi and Kolwezi experienced significant economic growth, mainly due to the liberalization of mining activities in 2002 [30,74]. Therefore, their spatial configuration has undergone profound changes, in particular, due to the densification of buildings and uncontrolled built-up area. This has had a considerable negative impact on the vegetation cover [28].

The Fungurume zone, still rural in character, has recorded a regression rate of the natural cover of 71%, exceeding the Likasi zone (61.5%). This zone saw industrial mining activities accelerate towards the end of the 2000s with the re-establishment of one of the country's major mining companies, alongside the multitude of artisanal mining sites [29]. In addition to the diggers, the presence of other actors can be reported within the mining sites (carriers, traditional authorities, industries, etc.), whose complex combination amplifies rivalries, both for space and resources [75]. The installation of thousands of people in the middle of the forest has an impact in terms of destruction of ecosystems, pollution, deforestation, etc. It also leads to soil degradation and loss of biodiversity [4]. Our results confirm the findings of [29,30,63], attesting to the fragmentation of the natural landscape and forest cover loss in Kolwezi and Fungurume zones due to mining pressure.

The modest forest cover loss that the Likasi zone is experiencing is thought to be related to the attraction of populations to other areas, notably Fungurume, Kolwezi and Lubumbashi zones, where large mining companies are concentrated. Furthermore, the land cover dynamics observed in the Likasi zone attest to the conclusions of [76], confirming deforestation in this region. This situation does not spare the forest resources of the Lufira Biosphere Reserve, located on its periphery and where the deforestation rate exceeds the national average [77].

In the four zones, the low trends of regression in natural cover observed between 1979 and 2000 would be justified by the collapse of several agricultural and mining operations after Zairianization (the process of nationalizing foreign companies undertaken in 1973 in DR Congo) and the civil wars. The period between 2000 and 2010 saw the beginning of the rural exodus following the resumption of mining activities in the region [18]. Reference [78] reported an acceleration of deforestation in Katanga during this period due to anthropogenic pressure and mining activities before experiencing a slowdown caused by the economic crisis of 2008. The results of our study demonstrate this acceleration of landscape anthropization in the last decade (2010–2020). This situation is explained by the rural exodus as well as the intrinsic growth of the urban population coupled with the inappropriate management of built-up area expansion [62,74], a consequence of the emergence of mining activity after the economic crisis. Numerous studies illustrate these increased changes related to agriculture, built-up area expansion, charcoal production and mining activities in the sub-region at the expense of natural cover [17,63,65,68]. This is also consistent with the findings of [10] that both deforestation and forest degradation in the Congo Basin have significantly accelerated in recent years. The trends observed

around the main anthropization poles in southern Katanga confirm the findings of [79] that charcoal production and agriculture have led to a loss of forest cover in southern Africa in general. The disturbance index values in this study obtained confirm these trends. In Lubumbashi, ref. [17] recorded significantly lower disturbance values than our study due to the difference in spatial and temporal scale.

The trends observed after first-order Markov chain modeling indicate a progression of built-up areas and mining cover by 2090 at the expense of other land covers, which is more noticeable in Lubumbashi and Kolwezi areas than in the Likasi and Fungurume areas. The conclusions of [55] attest that the forest cover loss will continue in the Katangan Copperbelt in the predominantly mining areas and worsen in the predominantly agricultural areas lacking adequate infrastructure in the future. Similar trends were obtained by [60], where the expansion of savannah and agricultural areas is expected in the woodland located in the southern zone of the forest–savanna transition region in eastern Côte d'Ivoire.

### 4.3. Implications for the Management of Peri-Urban Landscapes

This study showed that the agglomerations of Southeastern Katanga had undergone significant built-up area expansion as a result of their uncontrolled expansion accompanied by a strong regression of vegetation cover, especially in peri-urban areas. It should be noted that people living in peri-urban areas are trying to survive in a fragile economic context with little concern for the sustainability of the plant formations present. However, these plant formations will become very important in the near future for sustainable development. However, Ref. [17] reported high plant diversity in peri-urban areas, and green spaces in these areas provide good quality ecosystem services compared to those in urban areas [80]. The importance of vegetation is undeniable: it is a source of well-being and pleasure, and its soothing power contributes to the reduction of certain urban environmental problems such as water and air pollution or urban heat islands [81,82]. Good zoning followed by legal security of the green spaces to be preserved in the urban and peri-urban areas of Southeastern Katanga would be a good solution to satisfy the increasing need for nature in the vicinity of the city dwellers. In order to support the principle of rational use of space, building densification and the production of housing development according to the real needs of the population to avoid de-densification should be favored. The process of de-densification of the built environment generally leads to a waste of the resource "geographical space", which is considered to be limited. A trend towards land saturation was observed in West African countries, in areas with high population density but without land-use planning [83].

The study also showed that natural vegetation cover is being destroyed to meet agricultural and energy needs. In the lack of development regulations in agricultural areas (urban and peri-urban agriculture) coupled with support for farmers, and without diversification of energy sources, the worst is yet to come. The demographic pressure on peri-urban areas will continue. Innumerable harms of the deforestation of forest ecosystems can be observed. For example, there is a progressive disappearance of many non-wood resources, gathering products such as honey, mushrooms, but also caterpillars due to the selective cutting of host plants, etc. [84]. This deterioration also leads to a significant reduction in game fauna and a progressive deficiency of animal proteins that the village population obtains from hunting (game) and gathering (caterpillars) products [85]. Another consequence is the reduction of albedo, which in turn reduces the number of rainy days [86]. The preservation of existing forest patches is an issue that could lead to the imposition of land-use planning and urban development consistent with nature conservation [5,14]. Another way to preserve forests is to encourage community forestry around cities.

Additionally, observation of land cover maps reveals that building densification, which involves the anthropization of the peri-urban environment, is taking place on land set aside for green spaces to absorb the urban population in search of housing in the Southeastern Katanga. Most of the urban trees (exotic and local species) that were cultivated to improve the greenery and protect the urban environment have been destroyed or degraded due

to the rapid growth of the cities' population. In the same sense, our results revealed that they are non-existent in unplanned neighborhoods whose creation is accompanied by the systematic destruction of the pre-existing vegetation cover; this is in line with the observations of [87]. Reforestation solutions should be considered, both within and around cities. However, this process should avoid exotic species, some of which are potentially invasive and may further degrade ecosystems. In this context and based on seedling biomass, native species such as *Brachystegia spiciformis* Benth, *Combretum collinum* Fresen and *Pterocarpus tinctorius* Welw are the most productive potential candidates for reforestation of degraded miombo woodland [88].

## 5. Conclusions

This study aimed to show the stages of landscape anthropization in southeastern Katanga in a mining context. Our results show a remarkable land cover change, materialized through a regression of natural cover in favor of anthropogenic land cover. The loss of natural cover is considered too low in 1979, low between 1984 and 2000, average in 2005 and 2010 and accelerates in 2015 and especially in 2020. This trend was observed in all four zones. The inverse situation was observed for the built-up area, which increased remarkably over the same period and more markedly in Lubumbashi and Kolwezi regions compared to the Fungurume and Likasi regions. Agricultural and energy production increased significantly in all four zones. In terms of the landscape anthropization stage, the landscape will remain dominated by agriculture and energy production in 2090 for the Fungurume and Likasi zones. However, a transition materialized by the replacement of agricultural and energy production spaces, the matrix of the landscape by built-up areas will constitute the new dominant matrix of the landscape in 2060 and 2075, respectively, for the Lubumbashi and Kolwezi zones. The landscape degradation trends observed in this study and the resulting ecological consequences are to be feared in the absence of an adequate land-use planning policy.

This policy should (1) set up a way of preserving the few remaining forest patches around the cities of Lubumbashi and Kolwezi within a radius of 25 km, (2) define protection zones around Likasi and Kolwezi as there are still patches of forest around these two cities in the southern parts and (3) encourage community forestry in the villages around the urban areas

**Author Contributions:** H.K.M.: writing—original draft, D.-D.N.N.: writing—review, F.M.: writing—review, S.W.: writing—review, K.R.S.: writing—review, S.C.K.: writing—review, F.M.K.: writing—review, J.-F.B.: writing—review, J.B.: supervision and writing—original draft, Y.U.S.: supervision and writing—original draft. All authors have read and agreed to the published version of the manuscript.

**Funding:** The research was funded by the project CHARLU (ARES-CCD, Belgium).

**Institutional Review Board Statement:** Not applicable.

**Informed Consent Statement:** Not applicable.

**Data Availability Statement:** The data presented in this study are available on request from the corresponding author.

**Conflicts of Interest:** The authors declare no conflict of interest.

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
