# Peer review of "Quantification and Simulation of Landscape Anthropization around the Mining Agglomerations of Southeastern Katanga (DR Congo) between 1979 and 2090"

_land, doi:10.3390/land11060850_

Round 1

Reviewer 1 Report

Quantification and simulation of landscape anthropization around the mining agglomerations of southeastern Katanga (DR Congo) between 1979 and 2090

Line 51. Change the “indeed” for another word, because you recently use it (Line 48).

Line 60. You are constantly employing “indeed”. Try another word.

Line 64. Attend in the same way to the Line 60.

Line 186. Try to put the cite 27 after the word “model” (Bogaert an others model [27])

Line 195. Change the word “study” for “work” in “… referred in this study as…”.

Line 239. Why do you suggest as statistically reliable your accuracy values? I suggest you to answer this question by citing something about Kappa. You need to reaffirm your findings with certainty.

Line 377. Sometimes you write “miombo” in cursives, and sometimes you don´t. Homogenize that.

Line 379. Attend this line in the same way like Line 60.

Line 407. You are being too repetitive with the “indeed” word during all the manuscript.

Line 488. Who are the “several authors” that confirms a trend towards a land saturation? Cite them.

Line 518. The first sentence is not necessary (from Line 518 to Line 521, how far do you put the point after “tropical mining”). Try to express the most relevant findings.

Author Response

Thank you very much for your pertinent remarks, please find answers in the attachment

Reviewer 2 Report

Dear Authors,

Congratulations for your manuscript, it reflects a thorough and accurate work. I suggest the article for minor modifications - please see my comments and highlighted parts in the attached PDF.

Author Response

(The authors gave the same response as above.)

Reviewer 3 Report

Dear Authors,

I am sending a few remarks to yours article (see the attachment). In my opinion it is a very important research problem

Best regards

Author Response

(The authors gave the same response as above.)

Reviewer 4 Report

The article is certainly well written, and I recommend its acceptance for publication after minor revision. 

Minor revisions:

  1. The keyword part must be rewritten, and the author needs to select some representative words, such as southeastern Katanga, mining agglomerations, landscape anthropization, land use/cover change, etc.
  2. The format of the reference was not prepared in accordance with the requirement of the journal.

Author Response

(The authors gave the same response as above.)
